# Antilipidemic and Hepatoprotective Effects of Ethanol Extract of *Justicia spicigera* in Streptozotocin Diabetic Rats

**DOI:** 10.3390/nu14091946

**Published:** 2022-05-06

**Authors:** Marina Murillo-Villicaña, Ruth Noriega-Cisneros, Donovan J. Peña-Montes, Maribel Huerta-Cervantes, Asdrubal Aguilera-Méndez, Christian Cortés-Rojo, Rafael Salgado-Garciglia, Rocío Montoya-Pérez, Héctor Riveros-Rosas, Alfredo Saavedra-Molina

**Affiliations:** 1Instituto de Investigaciones Químico Biológicas, Universidad Michoacana de San Nicolás de Hidalgo, Morelia 58030, Mexico; 0850421k@umich.mx (M.M.-V.); 0618853j@umich.mx (D.J.P.-M.); marzy112@yahoo.com.mx (M.H.-C.); amendez@umich.mx (A.A.-M.); christian.cortes@umich.mx (C.C.-R.); rafael.salgado@umich.mx (R.S.-G.); rocio.montoya@umich.mx (R.M.-P.); 2Facultad de Enfermería, Universidad Michoacana de San Nicolás de Hidalgo, Morelia 58030, Mexico; ruth.noriega@umich.mx; 3Departamento de Bioquímica, Facultad de Medicina, Universidad Nacional Autónoma de México, Avenida Universidad 3000, Cd. Universitaria, Ciudad de Mexico 04510, Mexico; hriveros@unam.mx

**Keywords:** antioxidant, bioactive compounds, diabetes, *Justicia spicigera*, liver, streptozotocin

## Abstract

Oxidative stress is a factor that contributes to the development of complications in diabetes; however, its effects can be counteracted using exogenous antioxidants that are found in some plants, which is why people turn to traditional medicines in the search for therapeutic treatment. *Justicia spicigera* has been demonstrated to have the capacity to reduce glycemic levels; however, its effects on non-insulin-dependent organs such as the liver have not been reported. During 30 days of administration of *Justicia spicigera* ethanol extract, the blood glucose and weight of rats were measured every 5 days. Once the treatment was concluded, the rats were sacrificed. Corporal weight, blood glucose, cholesterol, very-low-density lipoprotein (VLDL), triglycerides, total lipids, and liver profile were reduced in the diabetic condition and normalized with the application of ethanol extract from *J. spicigera* (EJS). Additionally, there was a significant increase in catalase and superoxide dismutase activity in the control diabetic rats, a decrease in their activity with the extract administration, and no effect on normoglycemic rats. In conclusion, EJS is considered to be capable of reducing oxidative stress by maintaining diminished lipid and liver function profiles in male Wistar rats with streptozotocin-induced diabetes.

## 1. Introduction

Diabetes mellitus (DM) is a heterogeneous set of multifactorial pathogenetic syndromes with a common nexus of metabolic disorder, mainly chronic hyperglycemia and alterations in lipid and protein metabolism. Glucose transportation through the plasma membrane of mammalian cells is one of the most important events of nutrient transport, since this monosaccharide has a central role in metabolism and cell homeostasis. To transport glucose inside the cells, the organs in the body have different GLUT proteins, and they are divided into two groups. One group is insulin-dependent, including skeletal muscle, adipose tissue, and heart tissue, which have GLUT4 transporters [1]. The other group is tissues that do not depend on insulin to transport glucose inside them, such as the brain, kidney, and erythrocytes, and the epithelial cells of the intestine and liver [1].

The liver performs several biochemical functions of synthesis and excretion, so there is no test that can define the state of total liver function. The National Academy of Clinical Biochemistry and the American Association for the Study of Liver Diseases recommend a specific panel of tests to be used in the initial evaluation of patients with known or suspected liver disease, designated as the liver function profile, which is composed of the following analyses: total proteins, albumin, aspartate aminotransferase (AST), alanine aminotransferase (ALT), alkaline phosphatase (AP), total bilirubin, and direct and indirect bilirubin [2]. As an example, liver necrosis transaminases, AST, and ALT are sensitive and specific for hepatocytes, as well as for total, direct, and indirect bilirubin [3]. Liver AP is found on the canalicular surface and is therefore a marker of biliary dysfunction [4].

In patients with diabetes, the prevalence of non-alcoholic hepatic steatosis or non-alcoholic fatty liver disease (NAFLD) is 50–75% because the liver is a key organ that contributes to the development of insulin resistance and type 2 diabetes mellitus [5,6]. Therefore, dyslipidemia is a major side effect of diabetes [6]. Mitochondria, the major producers of reactive oxygen species (ROS) in the electron transport chain, also contribute to oxidative and nitrosative stresses when ROS are generated and lipid peroxidation occurs in diabetes [7,8]. Similarly, the mitochondrial glutathione pool is affected in diabetes due to oxidative stress [7]. 

Plants are among the most promising sources for discovering new antioxidant agents [9]. Since antiquity, humans have consumed the seeds, roots, stems, flowers, and fruits of plants to alleviate disorders due to their effectiveness in healing, their availability, and their low cost. Polyphenols are metabolites characterized by the attachment of one or more hydroxyl groups to one or more aromatic rings [10]. Among polyphenols, flavonoids represent a wide variety of metabolites [11]. including flavonoid glycosides, which are mainly found as their 3- or 7-O-glycosides [12]. In particular, flavonoids are considered as potential antidiabetic agents because they have multiple actions that are both hypoglycemic (insulinomimetic action) and antihyperglycemic (insulin secretagogue) [13]. Previous studies have shown that extracts of *Justicia spicigera* have antioxidant properties [14]. because they contain significant concentrations of flavonoids, mainly kaempferol glycosides, the most important being: kaempferitrin (kaempferol-3,7-dirhamnoside) and astragalin (kaempferol-3-β-D-glucopyranoside) [15], of which their antidiabetic properties have been demonstrated [16,17], and anticancer [18,19]. Consequently, we aimed to study the hepatic function and lipid profiles of an insulin-independent organ and analyze the antioxidant activity of *Justicia spicigera* in liver mitochondria of Wistar rats with streptozotocin-induced diabetes.

## 2. Materials and Methods

### 2.1. Plant Material and Extraction

*Justicia spicigera* plants were collected in the spring, during April and May 2021, in Morelia, Michoacán, Mexico, from the greenhouse of the Instituto de Investigaciones Químico Biológicas of the Universidad Michoacana de San Nicolás de Hidalgo. Briefly, *Justicia spicigera* leaves were collected and the fresh plant material was prepared by maceration with ethanol to obtain higher yields in the recovery of flavonoids [20,21], for 6 days at 4 °C, with 10 mL of solvent added per 1 g of plant material, for effective dissolution and extraction of polyphenolic compounds. After filtration, the extracts were evaporated to dryness in a rotary evaporator with reduced pressure at 55 °C and dissolved in DMSO (5%) to a final concentration of 100 mg/mL. The extracts were stored at 4 °C until use.

### 2.2. In Vitro Antioxidant Assays

#### 2.2.1. DPPH Radical Scavenging Assay

The DPPH^•^ (2,2-diphenyl-1-picrylhydrazyl) radical scavenging activity of the *J. spicigera* extract (100 mg/mL) was determined according to Lee et al. [22]. In brief, 0.1 mL of extract was made up to 1 mL with deionized water and mixed with 1 mL of DPPH solution (0.2 mM in absolute ethanol). Next, samples were incubated for 30 min in the dark at room temperature. A positive control was prepared with ascorbic acid (0.3 mg/mL). Absorbance was measured spectrophotometrically at 517 nm in a Perkin Elmer Lambda 18 UV-VIS spectrophotometer. The percentage of radical scavenging activity was calculated using the following formula: % RSA = ((Abs 517 control − Abs517 sample)/Abs517 control) × 100

#### 2.2.2. Antioxidant Capacity Assay by Phosphomolybdenum

Antioxidant capacity by phosphomolybdenum was determined according to Prieto et al. [23]. For this assay, 0.1 mL of extract was mixed with 0.2 mL of deionized water, then 3 mL of the reactive phosphomolybdenum solution (0.6 M of H_2_SO_4_, 28 mM of Na_2_HPO_3_, and 4 mM of ammonium molybdate) was added and mixed. Then the reaction mix was incubated for 90 min at 95 °C. A positive control was prepared with ascorbic acid (0.3 mg/mL). Next, the reaction mix was cooled to room temperature, and absorbance was recorded at 695 nm in a Perkin Elmer Lambda 18 UV-VIS spectrophotometer. Antioxidant capacity of the extract was calculated as follows:% AC = ((Abs 695 sample/Abs 695 control)) × 100

#### 2.2.3. Reducing-Power Assay

The reducing-power activity of the *J. spicigera* extract (100 mg/mL) was assayed according to Cell Biolabs [24]. Briefly, 0.1 mL of extract was made up to 1 mL with deionized water. Next, 2.5 mL of phosphate buffer (0.2 M, pH 6.6) and 2.5 mL of 1% (*w*/*v*) potassium ferrocyanide were added and thoroughly mixed. Next, the reaction mix was incubated for 20 min at 50 °C. Then 1.5 mL of 10% (*w*/*v*) trichloroacetic acid was added, and the mix was centrifuged for 10 min at 3000 rpm. Finally, 2.5 mL of the supernatant was mixed with 2.5 mL of deionized water and 0.5 mL of 0.1% FeCl_3_. A positive control was prepared with ascorbic acid (0.3 mg/mL). The absorbance of the final reaction was measured at 700 nm in a Perkin Elmer Lambda 18 UV–VIS spectrophotometer. The reducing-power activity of the extract was calculated as follows:% Reducing power = (Abs 700 sample × 100)/Abs 700 control

#### 2.2.4. Antilipid Peroxidation Assay

The antilipid peroxidation assay is a modified thiobarbituric acid reactive substances (TBARS) assay for measuring lipid peroxidation, as described by Ohkawa et al. [25]. First, 0.5 mL of the *J*. *spicigera* extract (100 mg/mL) was made up to 1 mL with deionized water, 5 µL of 7 mM FeSO_4_ was added to induce lipid peroxidation, and the mixture was incubated for 30 min. Then, 1.5 mL of 20% (*v*/*v*) acetic acid (pH 3.5 adjusted with NaOH), 1.5 mL of 0.8% (*w*/*v*) thiobarbituric acid in 1.1% (*w*/*v*) sodium dodecyl sulfate, and 0.5 mL 20% (*w*/*v*) trichloroacetic acid were added, and the mixture was incubated in a boiling-water bath for 60 min, and centrifuged at 5000 rpm for 5 min. Ascorbic acid was employed as a positive control. Absorbance was measured at 532 nm in a Perkin Elmer Lambda 18 UV–VIS spectrophotometer. The percentage of antilipid peroxidation of the extract was calculated as follows:% Antilipid peroxidation = (Abs 532 sample × 100)/Abs 532 control

### 2.3. Animals

The studied animals were male Wistar rats (90 days old, 327–373 g). They were housed under standard laboratory conditions and maintained at room temperature in a room with a 12 h light/dark cycle, and fed a standard rodent diet and purified water ad libitum. We followed the recommendations of the regulatory standard for the use of animals issued by SAGARPA in the federal regulations for the use and care of animals (NOM-062-ZOO-1999). All protocols were approved by the Institutional Committee for the Use of Animals of the Universidad Michoacana de San Nicolás de Hidalgo (# 09/2018).

### 2.4. In Vivo Study

#### 2.4.1. Diabetes Induction

Diabetes was induced in overnight fasted rats by single intraperitoneal administration of streptozotocin (STZ) (50 mg/kg body weight) dissolved in fresh citrate buffer (pH 4.5). Control rats were injected with citrate buffer alone. Five days after induction, glucose levels were determined to confirm diabetes, and levels >300 mg/dL were considered for the study.

#### 2.4.2. Experimental Protocol

Rats were randomly divided into 4 groups: group I (normoglycemic + DMSO 5%) of 8 rats, group II (diabetic + DMSO 5%) of 5 rats, group III (normoglycemic + *J. spicigera* extract) of 6 rats, and group IV (diabetic + *J. spicigera* extract) of 5 rats. The ethanolic extract of *Justicia spicigera* (100 mg/mL) was administered at a dose of 100 mg/kg by oral gavage, the dose was obtained according to the results reported by Ortiz-Andrade et al. [26]. The treatment was continued daily for 30 days.

#### 2.4.3. Blood Glucose and Body Weight Determination

Blood glucose concentration was estimated by the enzymatic glucose oxidase method using a commercial glucometer (Accu-Chek Active, Roche) through tail tip puncture. Glucose estimation was started just before extract administration, and was done every 5 days for 30 days. Animal weight was recorded during the 30 days.

#### 2.4.4. Evaluation of Biochemical Parameters

At 30 days of treatment, the animals were fasted overnight and sacrificed by decapitation. Blood samples were obtained in BD Vacutainer^®^ dry tubes with coagulation activator, and serum was separated through centrifugation at 3500 rpm for 5 min for biochemical estimations. The lipid profile (total cholesterol, high-density cholesterol (HDL), low-density cholesterol (LDL), very-low-density lipoprotein (VLDL), triglycerides, total lipids, and atherogenic index) and the liver profile (total proteins, total bilirubin, direct bilirubin, indirect bilirubin, alkaline phosphatase, gamma glutamyl transferase, aspartate amino transferase, and alanine amino transferase) were determined in serum using DRI-CHEM NX500i dry chemistry equipment.

### 2.5. Mitochondrial Isolation

Mitochondria were isolated from the livers of male Wistar rats according to Hogeboom (1948) with some modifications [27]. The liver was cut into small pieces in a solution (220 mM mannitol, 70 mM sucrose, 2 mM MOPS, 1 mM EGTA, pH 7.4) at 4 °C. The suspension was homogenized and subjected to differential 2000 rpm centrifugation in a J2-MC device (Beckman) for 10 min at 4 °C, and the supernatant was centrifuged at 7500 rpm for 10 min. The last pellet was resuspended in a solution (220 mM mannitol, 70 mM sucrose, 2 mM MOPS, pH 7.4), and centrifuged at 9000 rpm for 10 min. Protein concentration was determined by the modified Biuret method [28].

### 2.6. Superoxide Dismutase (SOD) Determination in Liver Mitochondria

SOD activity, expressed in U/mg protein, was determined using a commercial analysis kit (Sigma-Aldrich, St. Louis, MO, USA). Readings were taken at 450 nm using a Multiskan Go microplate reader (Thermo Fisher Scientific, Vantaa, Finland).

### 2.7. Catalase (CAT) Activity in Liver Homogenate

Catalase enzyme activity in tissue was assayed, following the procedure of Jeulin et al. [29], by measuring the conversion of hydrogen peroxide to oxygen with a Clark-type electrode connected to a YSI 5300A Biological Oxygen Monitor (Yellow Springs, OH, USA).

### 2.8. Statistical Analysis

Results are expressed as mean ± standard error (SE). Statistical analyses were performed with one-way or two-way analysis of variance (ANOVA), with calculations done using GraphPad Prism (version 7) (GraphPad Software, San Diego, CA, USA). *p* < 0.05 was considered statistically significant.

## 3. Results

### 3.1. Evaluation of In Vitro Antioxidant Assays

The in vitro antioxidant assay results are presented in Table 1. DPPH^•^ and phosphomolybdate reduction results show 100% antioxidant activity of *J. spicigera* ethanol extract (100 mg/mL), the same value obtained for the ascorbic acid used as control, while the results of antilipid peroxidation and Fe-reduced assay (reducing power activity) show 78 and 25% activity, respectively, compared to 100% for ascorbic acid.

### 3.2. Evaluation of Body and Liver Weight in Experimental Animals with Hyperglycemia

Experimental induction of diabetes was confirmed by elevated blood glucose levels (Figure 1A), decreased body weight (Figure 1B), and signs of polyphagia and polydipsia in diabetic groups (data not shown). The normoglycemic control group maintained normal glycemic levels of throughout treatment, starting at 80.5 ± 1.3 mg/dL glucose, latter 75.6 ± 2.6 mg/dL at 15 days, and 86.4 ± 3.4 mg/dL at 30 days after the first measurement. Treatment with *J. spicigera* ethanol extract for 30 days did not change blood glucose levels in the treated normoglycemic group, which presented glycemic levels of 81 ± 3 mg/dL at the beginning of treatment, 79 ± 1 mg/dL at the halfway point of treatment, and 74 ± 2 at the end of treatment; while glucose levels decreased significantly (*p* < 0.05) in the treated diabetic group that showed a glycemic level of 351 ± 43 mg/dL at the beginning of treatment, 266 ± 46 mg/dL, 15 days after the start, and 219 ± 39 mg/dL at the end of treatment, compared to the untreated diabetic control group, which presented elevated blood glucose levels throughout the experiment with 408 ± 19.1 mg/dL at the beginning, 438.7 ± 19 at the middle of treatment and a significant increase in glycemic levels of 427.3 ± 11.10 mg/dL at the end of treatment. Changes in body weight were recorded in grams (g) at the beginning and end of treatment. Figure 1B shows that the normoglycemic control group gained weight throughout the treatment, with an initial weight of 371 g ± 5 reaching 380 ± 8 g (2% weight gain). The normoglycemic group treated with the extract also gained weight, starting at 380 ± 9 g and reaching 396 ± 9 g at the end of treatment, a weight gain of 4%. The diabetic control and treated groups had significant (*p* < 0.05) decreases in body weight; the former had an initial weight of 373 ± 8 g and a final weight of 314 ± 12 g, showing a 16% decrease in body weight, while the latter started at 327 ± 13 g and ended at 293 ± 28 g, presenting a 10% decrease in body weight. The observed weight loss in the diabetic groups indicates good establishment of the diabetic model.

The weight of the liver obtained in the control group did not show significant differences when compared with the diabetic group treated with the extract; in the same way, there were no significant differences between the normoglycemic groups (Table 2).

### 3.3. Lipid Profile Evaluation

The effects of *J. spicigera* ethanolic extract on the lipid profile are shown in Table 3. The results obtained for serum HDL cholesterol from diabetic and normoglycemic rats showed no significant changes with administration of the extract. The same results were obtained for both groups treated with the extract. In relation to LDL cholesterol, the diabetic group exhibited significantly lower levels compared to the normoglycemic control group (*p* < 0.05); unexpectedly, treatment with *J. spicigera* ethanol extract lowered LDL cholesterol levels in normoglycemic animals (*p* < 0.05), but increased LDL cholesterol levels in diabetic animals (*p* < 0.05). On the other hand, VLDL cholesterol levels showed higher values in the diabetic group compared to the normoglycemic group (*p* < 0.05), and treatment with *J. spicigera* ethanol extract reduced VLDL cholesterol values in both groups (*p* < 0.05). Triglycerides exhibited higher values in the diabetic group compared to the normoglycemic group (*p* < 0.05), and both were significantly reduced (*p* < 0.05) in control glycemic and diabetic groups treated with the extract. The same significant results (*p* < 0.05) were obtained in the assay of total lipid content.

### 3.4. Liver-Profile Evaluation

#### 3.4.1. Total Proteins

The effects of *J. spicigera* ethanol extract on the liver profile are presented in Table 4 and Figure 2. The results show a high serum total protein content in the diabetic control group compared with the normoglycemic control group; however, the diabetic group treated with the extract showed reduced serum total protein compared with the diabetic control group without treatment, but there were no differences between the normoglycemic groups.

#### 3.4.2. Total Bilirubin

Total bilirubin in the normoglycemic group treated with the extract was reduced compared to the untreated normoglycemic group. The same results were obtained for the diabetic group treated with the extract. There were no significant differences in direct bilirubin for the normoglycemic groups; however, there was a decrease in the diabetic group treated with the extract compared to the diabetic group without the extract. In the normoglycemic control group treated with *J. spicigera* ethanol extract, significantly reduced indirect bilirubin values were obtained compared with the untreated control group (*p* < 0.05). In the diabetic group treated with the extract, there was a tendency toward reduced levels of indirect bilirubin compared with the untreated diabetic group.

### 3.5. Serum Liver Enzyme Activities

Alkaline phosphatase activity (Figure 2A) in the diabetic group treated with the extract showed a 31.15% reduction compared with the untreated diabetic group. Similarly, gamma glutamyl transpeptidase activity (Figure 2B) in the diabetic group treated with the extract showed a 19.11% reduction compared with the untreated diabetic group. With regard to alanine aminotransferase (Figure 2C) and aspartate aminotransferase (Figure 2D), the diabetic group treated with the extract showed a 73.24 and 68.26% reduction, respectively, compared with the untreated diabetic group.

### 3.6. Evaluation of Superoxide Dismutase and Catalase Activity

Figure 3A shows that treatment with ethanol extract of *J. spicigera* induced an increase in superoxide dismutase (SOD) activity in the normoglycemic group compared to the untreated group (*p* < 0.05). However, the diabetic group treated with the extract showed reduced SOD activity (*p* < 0.05). Catalase activity was the same in the liver homogenate from the treated and untreated normoglycemic groups, whereas catalase activity in the diabetic group treated with the extract was significantly reduced compared with the untreated diabetic control group (*p* < 0.05) (Figure 3B).

## 4. Discussion

Patients with DM have increased oxidative stress and inflammatory processes, which are greater in those who present with diabetes complications [30,31,32]. The liver plays a fundamental role in oxidative and detoxification processes. Therefore, to prevent or control the occurrence of complications, such as liver disease in patients with diabetes, the use of an antioxidant compound that complements treatment should be considered. There are reports on the presence of flavonoids, such as kaempferitrin and its bis-ramnoside, kaempferol, in the leaves of *J. spicigera* [33]. Kaempferol is a potent antioxidant that prevents oxidative damage of cells, lipids, and DNA [34], and has also shown hypoglycemic properties in in vitro and in vivo assays [35]. Studies of *J. spicigera* by García-Márquez et al. [36] reported that extracts obtained using solvents with greater polarity showed more effective radical scavenging activity than extracts obtained using solvents of lower polarity. Sepúlveda-Jimenez et al. [37] analyzed the aerial part of *J. spicigera* and observed that its methanolic extract had higher free-radical scavenging activity and a greater number of phenolic compounds and flavonoids.

In this work, we observed that ethanol extract of *J. spicigera* had antioxidant capacity similar to ascorbic acid (see Table 1), and therefore could provide protective effects against oxidative liver damage in diabetes when administered orally for 30 days. Similar antioxidant activity was reported by Awad et al. [14] with an ethanol extract of *J. spicigera*. 

On the other hand, treatment with ethanol extract of *J. spicigera* significantly decreased glycemic levels in streptozotocin-induced diabetic rats (Figure 1A). This is consistent with the results reported by Ortiz-Andrade et al. [26], who observed that *J. spicigera* ethanol extract had a hypoglycemic effect when administered to animals with experimental diabetes. This was probably due to the extract’s increased ability to assimilate glucose, which would explain the small increase in weight gain in normal rats (2% in untreated animals and 4% in treated animals) and the lower weight loss in rats with diabetes (Figure 1B), since these rats were probably able to take advantage of the ingested food, unlike the diabetic rats without the extract, which presented greater weight loss.

Patients with diabetes frequently present a combination of hyperglycemia and dyslipidemia [38,39]. This is because when insulin is lacking, the insulin-sensitive lipase enzyme in fat cells undergoes great activation. Thus, stored triglycerides are hydrolyzed, and large amounts of fatty acids and glycerol are released into the circulating blood. The fatty acids entering the hepatocytes, together with fatty acids derived from de novo lipogenesis, are used for the synthesis of triglycerides and other complex lipids [40]. The hormone-sensitive lipase catalyzes the hydrolysis of triglycerides in adipose tissue, which results in the release of fatty acids into the circulation. Normally, insulin suppresses this release and blocks the release of fatty acids. However, in insulin resistance states, insulin fails to suppress the release, which results in enhanced lipolysis and increased fatty acid flux to the non-esterified fatty acid plasma pool [41]. Sheweita et al. [42] reported that plasma levels of triglycerides, total cholesterol, LDL, and VLDL were increased in diabetic rats induced with STZ compared with control rats, while there was a decrease in HDL, thus maintaining an adequate model of diabetes.

Streptozotocin-induced diabetic rats showed an increased serum content of total lipids and triglycerides in comparison to control rats (Table 3). This agrees with previous published works reporting an increase in hepatic synthesis of triglycerides and subsequent hypertriglyceridemia in diabetic subjects [43]. Treatment with ethanol extract of *J. spicigera* in both the normoglycemic control group and streptozotocin-diabetic group showed a significant decrease in serum triglycerides and total lipids (Table 3), producing a hypolipidemic effect that was reflected in a decrease in the plasma atherogenic index. This improvement could be due to the presence of some component in the extract capable of normalizing triglyceride levels, and hence total lipids, in diabetes, as the results show a similar a total cholesterol content in all experimental groups, suggesting that *J. spicigera* ethanol extract does not participate in inhibiting cholesterol synthesis (Table 3). This is related to the results obtained when quantifying HDL cholesterol, where no significant changes were observed in any group (Table 3). On the other hand, a significant increase in VLDL cholesterol was observed in the untreated diabetic control group (Table 3), probably due to the excessive production of fatty acids in the liver that are secreted as VLDL components; this metabolic situation is also responsible for the increased hepatic synthesis of triglycerides and subsequent hypertriglyceridemia [34] observed in the control diabetic group (Table 3). 

The percentage of total liver weight was calculated based on body weight to denote the size of the liver after receiving treatment with ethanolic extract of *J. spicigera* (100 mg/mL). The total liver/body weight, we observed that the diabetic group treated with the extract showed increased liver weight in relation to body weight (Table 2); these results are consistent with those obtained by Noriega-Cisneros et al. [7], suggesting that an increase in liver size may be attributable to the administered extract acting at the liver level on lipid storage and mobilization, thus modifying blood lipids, as was already reported [44].

Because liver diseases are more frequent in the diabetic population [45], the present investigation was carried out to evaluate the potential protective effect of *J. spicigera* on this organ. The determination of total proteins and bilirubin allows us to directly identify changes in liver metabolic function, while the activity of liver enzymes in plasma is a reliable marker for assessing liver damage [46], including oxidative damage. Albumin is the main protein produced by the liver, and it can be altered when there is liver damage, a catabolic state, malnutrition, or loss of proteins; it is also responsible for transporting numerous endogenous substances such as bilirubin [47,48]. In our study, the groups treated with *J. spicigera* ethanol extract, both control and diabetic groups, showed decreased total protein levels. The liver secretes most of the proteins in blood plasma, including sex-hormone-binding globulin (SHBG). The circulating concentration of SHBG is associated with glucose metabolism, adiposity, and components of the metabolic syndrome [49]; therefore, liver disease can affect the plasma proteome [50]. This is probably what caused the increased serum levels of total proteins in the diabetic control group; however, further studies are required to corroborate the presence of this globulin in our study (Table 3). 

The increase in total and direct bilirubin occurs when there is some alteration at the hepatic level or of the bile ducts, while the increase in indirect bilirubin may reflect the presence of hemolysis. In our investigation, the levels of total, direct, and indirect bilirubin were reduced with the administration of the extract in the normoglycemic and diabetic groups (Table 4); direct bilirubin was unchanged in normoglycemic rats but diminished in the diabetic group. The latter suggests that *J. spicigera* extract offers protection from pathophysiological liver defects as observed in the streptozotocin-induced diabetic group.

Alkaline phosphatase can be present in organs other than the liver, so a supplementary test is required to confirm whether an increase in this enzyme comes from the biliary system or the liver [51,52], which is why its concomitant measurement with gamma glutamyl transferase is essential, since that enzyme comes almost exclusively from the liver [53,54]. The results obtained in this investigation indicated cholestasis and biliary dysfunction in the diabetic group without *J. spicigera* extract treatment, since both were increased; however, with administration of the extract, the levels were significantly improved in the diabetic condition and in normoglycemic rats (Table 4). These results are consistent with those obtained by Olayinka et al. [55]. 

The enzymes ALT and AST are the most commonly used indicators to assess the presence of liver necrosis [56]. Transaminases are sensitive but not very specific to hepatocyte damage, and ALT is more specific than AST, since it is not only found in the liver but also in skeletal and cardiac muscle, and increased levels of both enzymes have been reported [53,54,56,57]. These results are consistent with those obtained in our research for ALT and AST in diabetic rats without *J. spicigera* treatment. With extract administration, there was a significant decrease in those parameters in diabetic rats (Table 3); however, the results for LDH in the normoglycemic control group showed a significant increase compared to the control diabetic group, perhaps due to the variety of LDH isoenzymes that can be determined in serum, making this a nonspecific test. There was also a decrease in LDH activity in treated rats in both groups, indicating a positive effect with administration of the extract (Table 4). It is worth mentioning that there was a hepatotoxic effect of STZ. Ohaeri [58] observed that STZ-induced diabetic rats presented hepatic necrosis. Therefore, the increased activity of alkaline phosphatase, GGT, ALT, AST, and LDH in plasma could also be due to leakage of these enzymes from the hepatic cytosol into the bloodstream [59]. However, in this study, the ethanol extract of *J. spicigera* showed hepatoprotective activity by reducing the activity of these enzymes in plasma.

High levels of oxidative stress in diabetic animals are due to glucose autoxidation, protein glycation, lipid peroxidation, and low antioxidant enzyme activity [60]. High glucose in diabetes promotes higher production of reactive oxygen species in the presence of transition metal ions that cause oxidative damage to the lipids of cell membranes. However, the extent of damage seems to be specific, with the heart, liver, and kidney being more susceptible. The harmful effect of superoxide anion (O_2_^•−^) and hydroxyl radical (HO^•^) can be counteracted by antioxidant enzymes SOD and catalase (CAT). An increase in these enzymes has been indicated as a possible response mechanism in the early stages of diabetes [61]; however, intense long-term production of this radical exhausts the stimulation of enzymatic activity, since the reaction product can inhibit it [62]. This can occur at high CAT concentrations, since the cell uses the enzymes CAT and glutathione peroxidase for reduction of H_2_O_2_ [63]. Based on the above, we can deduce that the values obtained in our research for SOD in the diabetic control group (Figure 3A) may be due to inhibition of the enzyme by H_2_O_2_, since the observed CAT levels were increased in this group (Figure 3B). However, with the administration of *J. spicigera* ethanol extract in the diabetic group, the levels of both enzymes were reduced.

## 5. Conclusions

In conclusion, ethanol extract of *Justicia spicigera* exerted a protective effect on the livers of diabetic rats by reducing some characteristic symptoms of diabetes, such as hyperglycemia, body weight loss, serum triglycerides, and total serum lipids and by significantly reducing markers of hepatocyte injury such as gamma-glutamyl transpeptidase, alanine aminotransferase, aspartate aminotransferase, and alkaline phosphatase. However, since *J. spicigera* extract comprises a complex mixture of biological active compounds, some of its effects may potentially be undesirable, such as the reduced total serum protein content or reduced hepatic activity of catalase and superoxide dismutase observed in the streptozotocin-diabetic group. Further analysis of the individual effects of the main biological compounds contained in *J. spicigera* extract must be performed.

## Figures and Tables

**Figure 1 nutrients-14-01946-f001:**
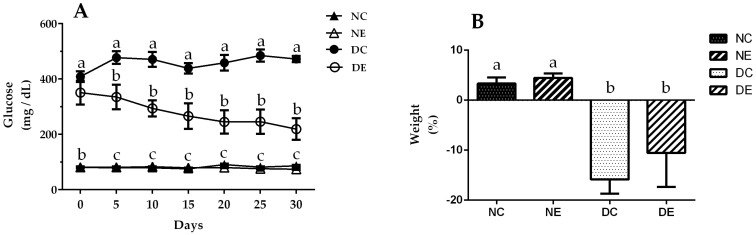
(**A**) Blood-glucose levels and (**B**) body weight during treatment with ethanolic extract of *Justicia spicigera* (100 mg/mL) for 30 days. Control (NC) and diabetic control (DC) groups were treated with DMSO; control administered (NE) and diabetic administered (DE) groups were treated with *J. spicigera* ethanol extract. Obtained values were analyzed by one-way ANOVA. Data represent mean ± SE (*n* = 5–8). Significant differences using Tukey’s multiple comparison test (*p* < 0.05) are indicated by different lowercase letters above each point; same letter indicates no significant differences.

**Figure 2 nutrients-14-01946-f002:**
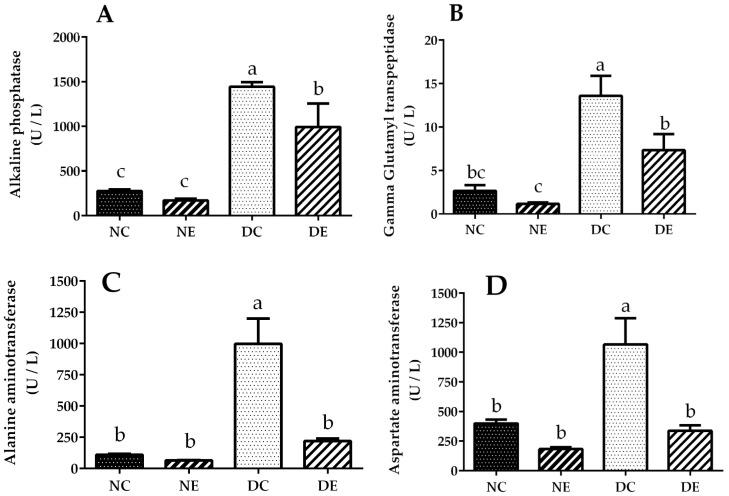
(**A**) Alkaline phosphatase, (**B**) gamma glutamyl transpeptidase, (**C**) alanine aminotransferase, and (**D**) aspartate aminotransferase activity at the end of treatment with ethanolic extract of *Justicia spicigera* (100 mg/mL) for 30 days. Control (NC) and diabetic control (DC) groups were treated with DMSO; control administered (NE) and diabetic administered (DE) groups were treated with *J. spicigera* ethanol extract (100 mg/mL). Values were analyzed by one-way ANOVA. Data represent mean ± SE (*n* = 5–8). Significant differences using Tukey’s multiple comparison test (*p* < 0.05) are indicated by different lowercase letters above each point; same letter indicates no significant differences.

**Figure 3 nutrients-14-01946-f003:**
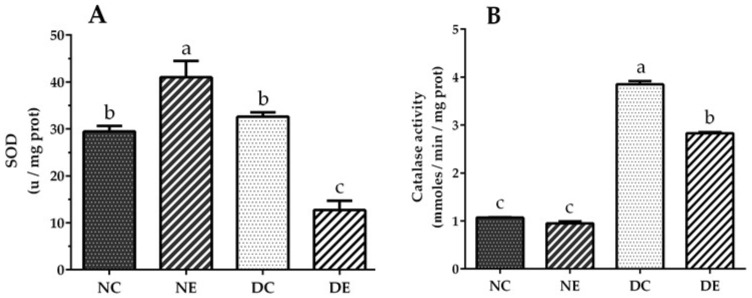
(**A**) Superoxide dismutase and (**B**) catalase activity from liver mitochondria and liver homogenate, respectively. Normoglycemic control group untreated (NC) and treated with *J. spicigera* ethanol extract (100 mg/mL) for 30 days (NE). DC, untreated diabetic group; DE, diabetic group treated with *J. spicigera* ethanol extract (100 mg/mL). Values were analyzed by one-way ANOVA. Results represent mean ± SE (*n* = 5–8). Significant differences using one-way ANOVA (*p* < 0.05) are indicated by different lowercase letters above each point; same letter indicates no significant differences.

**Table 1 nutrients-14-01946-t001:** Evaluation of in vitro antioxidant assays.

Assay	Control Ascorbic Acid (%)	*Justicia spicigera* Ethanol Extract (%)
DPPH^•^	100 ± 2.0 ^a^	105 ± 3.0 ^a^
Total antioxidant activity	100 ± 2.0 ^a^	101 ± 3.0 ^a^
Reducing power activity	100 ± 2.0 ^a^	25 ± 2.0 ^b^
Antilipid peroxidation	100 ± 2.0 ^a^	78 ± 4.0 ^b^

Obtained values of *Justicia spicigera* extract (100 mg/mL) were analyzed by one-way ANOVA. Data represent mean ± SD. (*n* = 3). Significant differences are denoted by superscript letters using Tukey’s multiple comparison test (*p* < 0.05); no significant differences are denoted by the same letter.

**Table 2 nutrients-14-01946-t002:** Liver weight percentage.

Group	Liver Weight (%)
NC	3.46 ± 0.18 ^bc^
NE	2.99 ± 0.07 ^c^
DC	3.86 ± 0.11 ^ab^
DE	4.42 ± 0.33 ^a^

Control (NC) and diabetic control (DC) groups were treated with DMSO; control administered (NE) and diabetic administered (DE) groups were treated with ethanol extract of *J. spicigera* (100 mg/mL). Obtained values were analyzed by one-way ANOVA. Data represent mean ± SE. (*n* = 5–8). Significant differences are denoted by superscript letters using Tukey’s multiple comparison test (*p* < 0.05); no significant differences are denoted by the same letter.

**Table 3 nutrients-14-01946-t003:** Lipid profile evaluation.

Lipid Profile			
Group	Total Cholesterol [mg/dL]	HDL Cholesterol [mg/dL]	LDL Cholesterol [mg/dL]
NC	88.33 ± 8.76 ^a^	36.67 ± 3.33 ^a^	19.80 ± 3.70 ^ab^
NE	75.40 ± 4.92 ^a^	45.00 ± 3.03 ^a^	14.56 ± 2.88 ^ab^
DC	73.67 ± 4.10 ^a^	42.00 ± 5.29 ^a^	7.40 ± 1.72 ^b^
DE	81.58 ± 2.86 ^a^	40.75 ± 0.48 ^a^	24.97 ± 3.85 ^a^
	VLDL-cholesterol [mg/dL]	Triglycerides [mg/dL]	Total lipids [mg/dL]
NC	31.87 ± 6.15 ^a^	110.00 ± 17.83 ^ab^	270.00 ± 15.34 ^b^
NE	15.84 ± 0.92 ^b^	79.22 ± 4.59 ^b^	278.33 ± 13.28 ^b^
DC	24.27 ± 4.62 ^ab^	172.33 ± 29.01 ^a^	440.70 ± 48.70 ^a^
DE	15.86 ± 1.48 ^b^	79.28 ± 7.41 ^b^	289.54 ± 12.04 ^b^
	**Atherogenic index of plasma**		
NC	0.48		
NE	0.25		
DC	0.61		
DE	0.29		

Control (NC) and diabetic control (DC) groups were treated with DMSO; control administered (NE) and diabetic administered (DE) groups were treated with ethanol extract of *J. spicigera* (100 mg/mL). Obtained values were analyzed by one-way ANOVA. Data represent mean ± SE. (*n* = 5–8). Significant differences are denoted by superscript letters using Tukey’s multiple comparison test (*p* < 0.05); no significant differences are denoted by the same letter.

**Table 4 nutrients-14-01946-t004:** Liver-profile evaluation.

Group	Total Protein [g/dL]	Total Bilirubin [mg/dL]	Direct Bilirubin [mg/dL]	Indirect Bilirubin [mg/dL]
NC	6.82 ± 0.20 ^b^	0.65 ± 0.10 ^a^	0.15 ± 0.07 ^ab^	0.50 ± 0.07 ^a^
NE	6.55 ± 0.53 ^b^	0.34 ± 0.05 ^a^	0.12 ± 0.02 ^ab^	0.22 ± 0.04 ^a^
DC	7.87 ± 0.30 ^a^	0.63 ± 0.12 ^a^	0.20 ± 0.00 ^a^	0.43 ± 0.12 ^a^
DE	5.90 ± 0.21 ^b^	0.33 ± 0.06 ^a^	0.10 ± 0.00 ^b^	0.23 ± 0.06 ^a^

Control (NC) and diabetic control (DC) groups were treated with DMSO; control administered (NE) and diabetic administered (DE) groups were treated with ethanol extract of *J. spicigera* (100 mg/mL). Obtained values were analyzed by one-way ANOVA. Data represent mean ± SE. (*n* = 5–8). Significant differences are denoted by superscript letters using Tukey’s multiple comparison test (*p* < 0.05); no significant differences are denoted by the same letter.

## Data Availability

The data used to support the findings of this study are available from the corresponding author upon request.

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
