# Peer review of "Antilipidemic and Hepatoprotective Effects of Ethanol Extract of Justicia spicigera in Streptozotocin Diabetic Rats"

_nutrients, 2022, doi:10.3390/nu14091946_

Round 1

Reviewer 1 Report

Manuscript Number: nutrients-1676537

Comments to Author

This article is interesting and fits with the aims of Nutrients, however in my opinion it is incomplete in the present form. Several major integrations should be carried out to improve the manuscript.

My comments are:

1) Introduction: Please specify the rationale of using Justicia spicigera extract in streptozotocin diabetic rats. More information should be added on the biological/nutraceutical properties of this plant and its extracts.

2) Literature data about the main phytochemical components of Justicia spicigera leaves should be reported in the Introduction section.

3)  In my opinion, the main problem of the manuscript is the lack of information on the chemical composition of the Justicia spicigera leave extract used in the in vitro and in vivo assays. The chemical characterization of the main components of the ethanol extract should be reported in the manuscript in order to better understand the observed antioxidant activity in the different in vitro assays and in rats. Moreover, the knowledge of the main extract constituents (and potential derivatives from the rat metabolism) is essential to understand and discuss the protective effects observed in streptozotocin diabetic rats.

Author Response

REVIEWER RESPONSE LETTER

Reviewer 1:

1) Introduction: Please specify the rationale of using Justicia spicigera extract in streptozotocin diabetic rats. More information should be added on the biological/nutraceutical properties of this plant and its extracts.

RESPONSE:

 The authors appreciate your kind observation. Please revise that more information was added, including new references from line 64 to 74 in the final version.

2) Literature data about the main phytochemical components of Justicia spicigera leaves should be reported in the Introduction section.

RESPONSE:

We the authors appreciate your observation, kindly, please revise that more information was added, including new references, that from line 69 to 74 in the final version.

3)  In my opinion, the main problem of the manuscript is the lack of information on the chemical composition of the Justicia spicigera leave extract used in the in vitro and in vivo assays. The chemical characterization of the main components of the ethanol extract should be reported in the manuscript in order to better understand the observed antioxidant activity in the different in vitro assays and in rats. Moreover, the knowledge of the main extract constituents (and potential derivatives from the rat metabolism) is essential to understand and discuss the protective effects observed in streptozotocin diabetic rats.

RESPONSE:

We the authors appreciate your observation. Kindly please the information included from line 69 to 74 in relation to the chemical composition of the Justicia spicigera extract.

Reviewer 2 Report

The authors have done a great job with this manuscript.

I invite you to kindly revisit the following points for some minor changes.

Line 10: There seems to be an extra email left from the template.

Line 19 and 21: Perhaps the name of the plant should be in italic here.

Line 76-83: As previous literature demonstrates, this plant has been evaluated in various settings as an aqueous extract, ethanol/water extract, methanol extract etc. Perhaps the authors could consider including a brief comment here, describing why the extraction of bioactive compounds in ethanol was selected for this study. Also in this setting, it may be beneficial for the reader to clarify how the ration of ethanol:plant (10ml:1g) was selected.

Line 82: Kindly revisit the value “100mg/kg”. As it reads, this is the “stock solution” of the extract used for the in vitro assays which is dissolved in DMSO. Perhaps this ought to be 100mg/ml?

Line 114-125: Was there a positive control for this assay? If so, please include it.

Line 124: In response to the results presented further this calculation should also be %. Kindly consider revisiting.

Line 143: Perhaps the authors would consider including the age of the animals.

Line 143: Also, in relation to the results presented further, the animals’ starting weight was in all cases lower than 400g. Perhaps the authors would consider including this in brief here and maybe a range of initial weight would be helpful.

Line 163: Kindly consider including a brief note regarding the selection of the treatment’s concentration and please include here any information the authors have regarding the potential toxicity of this concentration.

  • Section of Materials and Methods: Kindly revisit all assays and elaborate on the concentration of the extract evaluated. As it reads, the authors have included in all assays that 0.1ml or 0.5ml of the extract was diluted for each evaluation protocol but the initial “stock solution” remains unclear. Please consider that it is essential for the reader to understand if an initial solution from dried material was made and then from that solution, the authors made further dilutions for each evaluation protocol, or if the dried material was weighed to make a separate extract solution for each evaluation protocol.
  • Section of Results (Line 226_Table 1): In relation to the previous comment, the results presented in Table 1 are referring to the ethanol extract. Please consider elaboration if this reflects the initial extract (1g plant/10ml ethanol), the dried material diluted in DMSO (100mg dried extract in ?? ml DMSO), or any other solution of the dried material made to obtain these values.

Table 1: It is crucial for the reader as well as for further research that the authors would kindly consider presenting the outcomes of these evaluations in relation to a specific concentration.

Line 239-247: Kindly consider including the values of starting weight and final weight for each treatment group in mean (SD) in the text here.

Line 260_Figure 1(A): The authors have recorded a clear decline in Glucose levels for the DE group vs DC group. Perhaps it would be beneficial for the reader to report the starting, 15-days and then 30-days mean(SD) values in the text as well.

Line 272: Kindly consider elaboration on what the liver % values represent. Is there a liver weight reduction/increase compared to the control group? Please clarify the calculations here before further discussing them (line 474).

Figures 2, 3 and 4: Kindly consider including the extract’s concentration in the footnote of the graphs and tables when appropriate as the authors have done in the other parts of the manuscript.

Line 379: There seems to be an extra word here. Please revisit the context of the sentence.

Results vs. Line 163: Kindly elaborate on the meaning of n=5-8 in comparison to the number of animals in each group presented in line 163. Please revisit and revise the results accordingly.

Kindly consider adding a list of abbreviations at the end of the manuscript as it is usually very beneficial for the reader. This is clearly a suggestion to improve the quality of this great manuscript, as the authors have already included most of the abbreviated forms in the text. A second screening for missing abbreviations is also advised.

Author Response

REVIEWER RESPONSE LETTER

Reviewer 2:

Line 10: There seems to be an extra email left from the template.

RESPONSE:

Kindly the authors appreciate your observation, which it was corrected.

Line 19 and 21: Perhaps the name of the plant should be in italic here.

RESPONSE:

Kindly the authors appreciate your observation, which it was corrected.

Line 76-83: As previous literature demonstrates, this plant has been evaluated in various settings as an aqueous extract, ethanol/water extract, methanol extract etc. Perhaps the authors could consider including a brief comment here, describing why the extraction of bioactive compounds in ethanol was selected for this study. Also in this setting, it may be beneficial for the reader to clarify how the ration of ethanol:plant (10ml:1g) was selected.

RESPONSE:

The authors appreciate you observation. Kindly please revise the new information added in relation to ethanol solvent in this study, which is in line 85 to 87 in the final version.

Line 82: Kindly revisit the value “100mg/kg”. As it reads, this is the “stock solution” of the extract used for the in vitro assays which is dissolved in DMSO. Perhaps this ought to be 100mg/ml?

RESPONSE:

The authors appreciate your observation which kindly please revise that was corrected and appears in line 89 in the final version.

Line 114-125: Was there a positive control for this assay? If so, please include it.

RESPONSE:

The authors appreciate your observation. Kindly please revise that it was corrected and appears in line 127 to 128 in the final version.

Line 124: In response to the results presented further this calculation should also be %. Kindly consider revisiting.

RESPONSE:

The authors appreciate your observation. Kindly please revise that it was corrected and appears in line 132 in the final version.

Line 143: Perhaps the authors would consider including the age of the animals.

RESPONSE:

The authors appreciate your observation. Kindly please revise that it was corrected and appears in line 152 in the final version.

Line 143: Also, in relation to the results presented further, the animals’ starting weight was in all cases lower than 400g. Perhaps the authors would consider including this in brief here and maybe a range of initial weight would be helpful.

RESPONSE:

The authors appreciate your observation. Kindly please revise that it was corrected and appears in line 152 in the final version.

Line 163: Kindly consider including a brief note regarding the selection of the treatment’s concentration and please include here any information the authors have regarding the potential toxicity of this concentration.

RESPONSE:

The authors appreciate your observation. Kindly please revise that it was added a new information and appears in line 176 in the final version.

  • Section of Materials and Methods: Kindly revisit all assays and elaborate on the concentration of the extract evaluated. As it reads, the authors have included in all assays that 0.1ml or 0.5ml of the extract was diluted for each evaluation protocol but the initial “stock solution” remains unclear. Please consider that it is essential for the reader to understand if an initial solution from dried material was made and then from that solution, the authors made further dilutions for each evaluation protocol, or if the dried material was weighed to make a separate extract solution for each evaluation protocol.

RESPONSE:

The authors appreciate your observation. Kindly please revise that information was added in the Materials and Methods section in lines 89, 97, 109, 121, 138, and 234 in Results section in the final version.

  • Section of Results (Line 226_Table 1): In relation to the previous comment, the results presented in Table 1 are referring to the ethanol extract. Please consider elaboration if this reflects the initial extract (1g plant/10ml ethanol), the dried material diluted in DMSO (100mg dried extract in ?? ml DMSO), or any other solution of the dried material made to obtain these values.

RESPONSE:

The authors appreciate your observation. Kindly please revise that it was added a correction and appears in Table 1 in the final version.

Table 1: It is crucial for the reader as well as for further research that the authors would kindly consider presenting the outcomes of these evaluations in relation to a specific concentration.

RESPONSE:

The authors appreciate your observation. Kindly please revise that it was added a correction and appears in Table 1 in the final version.

Line 239-247: Kindly consider including the values of starting weight and final weight for each treatment group in mean (SD) in the text here.

RESPONSE:

The authors appreciate your observation. Kindly please revise that it was added a correction and appears in text from lines 257 to 264 in the final version.

Line 260_Figure 1(A): The authors have recorded a clear decline in Glucose levels for the DE group vs DC group. Perhaps it would be beneficial for the reader to report the starting, 15-days and then 30-days mean (SD) values in the text as well.

RESPONSE:

The authors appreciate your observation. Kindly please revise that it was added a correction and appears in text from lines 244 to 256 in the final version.

Line 272: Kindly consider elaboration on what the liver % values represent. Is there a liver weight reduction/increase compared to the control group? Please clarify the calculations here before further discussing them (line 474).

RESPONSE:

The authors appreciate your observation. Kindly please revise that it was added a correction and appears in text from lines 485 to 487 in the final version.

Figures 2, 3 and 4: Kindly consider including the extract’s concentration in the footnote of the graphs and tables when appropriate as the authors have done in the other parts of the manuscript.

RESPONSE:

The authors appreciate your observation. Kindly please revise that it was added a correction and appears in text from lines 381, 419, but there is not figure 4 in the final version.

Line 379: There seems to be an extra word here. Please revisit the context of the sentence.

RESPONSE:

Kindly your suggestion was attended and erased, now in line 396.

Results vs. Line 163: Kindly elaborate on the meaning of n=5-8 in comparison to the number of animals in each group presented in line 163. Please revisit and revise the results accordingly.

RESPONSE:

The authors appreciate your observation. Kindly please revise that it was added a correction and appears in text from lines 173 to 176 in the final version.

Kindly consider adding a list of abbreviations at the end of the manuscript as it is usually very beneficial for the reader. This is clearly a suggestion to improve the quality of this great manuscript, as the authors have already included most of the abbreviated forms in the text. A second screening for missing abbreviations is also advised.

RESPONSE:

The authors appreciate your observation. Kindly please revise that it was added a section and appears at the end of the manuscript in lines 586 to 613 in the final version.

Round 2

Reviewer 1 Report

The manuscript has been improved as a result of the revision.